# Variability in Biological Activities of *Satureja montana* Subsp. *montana* and Subsp. *variegata* Based on Different Extraction Methods

**DOI:** 10.3390/antibiotics11091235

**Published:** 2022-09-11

**Authors:** Milica Aćimović, Olja Šovljanski, Lato Pezo, Vanja Travičić, Ana Tomić, Valtcho D. Zheljazkov, Gordana Ćetković, Jaroslava Švarc-Gajić, Tanja Brezo-Borjan, Ivana Sofrenić

**Affiliations:** 1Institute of Field and Vegetable Crops Novi Sad, 21000 Novi Sad, Serbia; 2Faculty of Technology Novi Sad, University of Novi Sad, Bulevar Cara Lazara 1, 21000 Novi Sad, Serbia; 3Institute of General and Physical Chemistry, Studentski trg 10–12, 11000 Belgrade, Serbia; 4Department of Crop and Soil Science, Oregon State University, Corvallis, OR 97331, USA; 5Faculty of Chemistry, University of Belgrade, Studentski trg 12–16, 11000 Belgrade, Serbia

**Keywords:** winter savory, essential oil, hydrolate, subcritical water extraction, ultrasound-assisted extraction, microwave-assisted extraction

## Abstract

Winter savory (*Satureja montana* L.) is a well-known spice and medicinal plant with a wide range of activities and applications. Two subspecies of *S. montana*, subsp. *montana* and subsp. *variegata*, were used for the preparation of seven different extracts: steam distillation (essential oil (EO) and hydrolate (HY)), subcritical water (SWE), ultrasound-assisted (UAE-MeOH and UAE-H_2_O), and microwave-assisted (MAE-MeOH and MAE-H_2_O) extraction. The obtained EOs, HYs, and extracts were used for an in vitro evaluation of the antioxidant activity (DPPH, ABTS, reducing power, and superoxide anion methods) and in vitro antimicrobial activity against *Bacillus cereus*, *Staphylococcus aureus*, *Enterococcus faecalis*, *Escherichia coli*, *Salmonella* Typhimurium, *Saccharomyces cerevisiae*, and *Candida albicans*. The antimicrobial screening was conducted using disk-diffusion assessment, minimal inhibitory concentration, time–kill kinetics modeling, and pharmacodynamic study of the biocide effect. The total phenolic content (TPC) was highest in EO, followed by SWE, MAE, and UAE, and the lowest was in HY. The highest antimicrobial activity shows EO and SWE for both varieties, while different UAE and MAE extracts have not exhibited antimicrobial activity. The natural antimicrobials in the *S. montana* extract samples obtained by green extraction methods, indicated the possibility of ecologically and economically better solutions for future in vivo application of the selected plant subspecies.

## 1. Introduction

The application of herbs, spices, and medicinal plants and their products in the food industry, as preservatives, aroma, and flavor additives, and in many other fields such as folk medicine and pharmacy, has a long history [1]. Furthermore, interest in natural antioxidants has intensified due to the concerns about the safety of synthetic antioxidants and the assumption that plant-derived compounds are safer and beneficial to human health [2]. Therefore, the phytochemical screening of herbal extracts leads to the acceptance of new prospective sources of natural chemical compounds such as antioxidants [3].

Genus *Satureja* (Lamiaceae family) compromises 30 species mainly distributed in the Mediterranean area [4]. Some of the species are endemic, while others are found in broader regions. *Satureja* species are annual or perennial shrubs found mostly on dry, sunny, and rocky terrains [5]. The plants from this family are characterized by intensive odor originating from the essential oil (EO), which accumulates in glandular trichomes on the surface of stems, leaves, calyces, and corollas [6]. *Satureja montana* L. *sensu lato*, commonly known as mountain or winter savory, is a perennial aromatic shrub that grows wild on mountain ranges in the Adriatic area, such as the Dinaric Alps and the Apennine Mountains, including Italy [7,8,9], Slovenia [10,11], Croatia [12,13,14], Bosnia and Herzegovina [15,16], Serbia [17], Montenegro [18,19], and Albania [20]. It also occurs in the Pyrenees, i.e., in France [21] and Spain [22]. The wide range of distribution, and the influence of microclimatic and edaphic conditions and crossover with other *Satureja* species, has led to high variability in the botanical characteristics and chemical composition within the populations. *S. montana* is a well-known plant used in the traditional medicines of many Mediterranean countries, as well as spice and flavoring agents. However, high biological potential makes them cognizable as significant raw materials for the pharmaceutical and food industries. Currently, the significant application is in traditional medicine to treat several ailments such as cramps, diarrhea, muscle pains, nausea, and infectious diseases [5,23,24]. Furthermore, antibacterial, antifungal, antioxidant, antispasmodic, antidiarrheal, carminative, digestive, antidiuretic, etc. properties have been reported [2,25,26]. Commonly, *S. montana* is used for EO production via the steam distillation process, whereas the by-products are occurring hydrolates (HYs). In the last two decades, they have attracted significant attention from scientists as a means to reduce waste and environmental pollution and increase the financial gain of EO production [27].

As one of the plant species that are rich in phytochemicals, *S. montana* has been studied as an antimicrobial agent in recent decades. As it is rich in volatile and nonvolatile compounds, this plant showed a strong antimicrobial effect against a wide range of different types of microorganisms. Briefly, many researchers implied that the stronger biocide effect of *S. montana* exists against gram-positive bacteria compared to gram-negative bacteria [28,29], but also different types of eukaryotic organisms can be inhibited when in contact with the plant oil or extracts [30,31,32]. Despite the fact that *Staphylococcus aureus* and *Bacillus cereus* have been repeatedly proven to be sensitive to bacteria on *S. montana*, the variable behavior of *Esherichia coli* was observed [18,28,29,30,31,33,34,35]. Furthermore, the biocide effect against *Enterococcus faecalis* was observed in the study [30,31]. Interestingly, only one study on the antimicrobial activity of *Satureja montana* involved *Salmonella* Typhimurium, and a positive outcome was observed (the inhibition effect was very high) [36]. Minimal or average antibacterial activity was reported against *P. aeruginosa* reference strains [30,31,33,35,37]. The inhibition effect on eukaryotic organisms was uniform throughout the scientific literature, showing the effect against two main representatives: *Saccharomyces cerevisiae* and *Candida albicans* [30,31,32]. However, the fungistatic to fungicide effects of EOs of *S. montana* using oil concentration in the range of 150 to 300 μg/mL against nine phytopathogenic fungi have also been reported [38]. Almost all of the mentioned studies refer to the testing of essential oil, and it can be concluded that the sample types (in view of different extraction (conventional and novel) as well as hydrolates as a by-product of essential oil production, were tested by individual researchers or were not tested at all.

Green technologies involve the use of new approaches for the extraction of important bioactive compounds from different sources, such as herbs, algae, and other organisms, leading to a reduction in environmental pollution and energy consumption, allowing the use of alternative solvents, and providing safe, high-quality extract/product. Moreover, these green extractions are fulfilling circular economy principles [39,40]. The high demand for herbal products necessitates the discovery of the most effective methods for the extraction of particular compounds from plants [41]. The extraction techniques can be divided into two groups: conventional (e.g., maceration, heat-assisted extraction, percolation, Soxhlet extraction) and non-conventional (microwave-assisted extraction, supercritical fluid extraction, subcritical water extraction, accelerated solvent extraction, ultrasound-assisted extraction) [42,43]. Conventional extraction techniques have some disadvantages, such as the thermal and hydrolytic degradation of bioactive compounds, the use of large quantities of organic solvents, and their residues in the final product [44]. In recent decades, non-conventional techniques have been recognized to have significant potential in the extraction process, requiring less time, energy, and solvents [42,45,46].

In addition to EO and HY, the following novel and green methods were involved in this study: microwave-assisted extraction (MAE), ultrasound-assisted extraction (UAE), and subcritical water extraction (SWE). MAE uses liquid solvents to extract the active compounds from a matrix causing migration of polar molecules in a generated time-varying electric field, resulting in a classification of compounds in the existing electric field. MAE typically results in a short extraction time and high extraction yield [41,46]. UAE uses the cavitation, thermal, and mechanical properties of ultrasound to induce cell wall disruption and degrade the plant matrix and improve the mass transfer of important compounds enabling easier access to solvent to the matrix [42,46]. SWE is a technique where water is used at temperatures and pressures above the critical point, resulting in relatively low viscosity and high diffusivity that increase diffusion and mass transfer leading to a reduction in extraction time [46].

Since there is no information about the different extraction methods of different varieties of *S. montana*, the present study aimed to evaluate and compare antioxidant and antimicrobial activities of two subspecies of *S. montana.* The extracts obtained by microwave-assisted, ultrasound-assisted, and subcritical water extraction were analyzed and compared with the EO and HY concerning their biological properties and bioactivity.

## 2. Results

### 2.1. Chemical Composition

#### 2.1.1. Volatile Compounds

The volatile fraction of the EO and HY analyzed by GC/FID and GC-MS is shown in Table 1. The EO of *S. montana* subsp. *montana* contained carvacrol (35.7), p-cymene (32.3%), and γ-terpinene (10.5%) as the main compounds, while *S. montana* subsp. *variegata* EO contained thymol (51.4%), carvacrol (13.1%) and p-cymene (10.7%). The HYs of both subspecies contained carvacrol as the dominant compound (91.6% and 96.4%, respectively).

#### 2.1.2. Total Phenolic Content

All of the samples were tested in view of total phenolic content (TPC), and the obtained results are presented in Figure 1. As expected, the greatest phenolic content has EOs. However, *S. montana* subsp. *variegata* contains a higher level of TPC compared to *S. montana* subsp. *montana*.

On the other hand, the highest TPC was observed for SWE extracts with a better score for *S. montana* subsp. *montana* compared with that of *S. montana* subsp. *variegata*. A slight difference was observed between the water- and methanol-based MAE samples, while a significant difference was found with the UAE samples, where using methanol as a solvent had better extraction efficiency compared to water in view of the total phenolic content. The presence of total phenolic compounds was the lowest in the case of both tested HYs. In summary, ranking of samples with respect to total phenolic content was: EO > SWE > MAE-MeOH > MAE-H_2_O > UAE-MeOH > UAE-H_2_O > HY.

### 2.2. In Vitro Antioxidant Activity

Figure 2 presents the antioxidant activity of the extracts, which was determined using four different assays: 2,2-diphenyl-1-picrylhydrazyl (DPPH) and 2,2-azino-bis-3-ethylbenzothiazoline-6 acid (ABTS), reducing power according, and superoxide anion (SOA). The obtained results indicate significant differences between the used assays for the antioxidant determination, which are explained in the following discussion.

### 2.3. In Vitro Antimicrobial Activity

The in vitro antimicrobial activity of *S. montana* was qualified by inhibition zone diameters using the disc diffusion method and quantified by determining the minimal inhibitory concentration (MIC) using the microdilution assay, as well as defining the time–kill kinetics of the antimicrobial effect using a pharmacodynamics assay. As shown in Figure 3, the spectrum of the antimicrobial activity varied between the investigated *S. montana* subspecies as well as the extraction methods. The obtained results suggest that Gram-positive bacteria were more sensitive to the effect of the tested samples, especially in the case of Eos and Hys. In contrast to these results, the HY of both subspecies revealed antibacterial activity only against cocci bacteria but was not effective against sporogenic *B. cereus*, Gram-negative bacteria, and fungi. Both tested yeasts were sensitive to the HY, while a higher inhibition zone was observed for *S. cerevisiae*. On the other hand, the tested SWE samples have revealed antibacterial activity against only *B. cereus*. Different UAE and MAE extracts have not exhibited antimicrobial activity, except methanolic UAE extract of *S. montana* subsp. *Montana*, which inhibited *S. cerevisiae* and *C. albicans*.

Among the controls, pure carvacrol, thymol, and p-cymene were tested as the predominant constituents in the tested Eos and Hys. Carvacrol presented the most effective antimicrobial capacity that expresses its activity against *S*. *aureus*, *E*. *faecalis*, *E*. *coli*, *S*. Typhimurium, *S*. *cerevisiae*, and *C*. *albicans* (inhibition zones range 19–31 mm) (Figure 3). Additionally, thymol has partially inhibited the growth of *P. aeruginosa*, *S*. Typhimurium, and *A. brasiliensis*, while the antimicrobial activity of p-cymene was observed against *B. cereus*, *S*. Typhimurium, *S. cerevisiae,* and *C. albicans*. The results suggest synergistic and combined activity of the dominant constituents in the *S. montana* extracts. This also indicated that the extracted ingredients were the main compounds responsible for biological activity, while their content in extracts was directly connected to the applied extraction technique.

Minimal inhibitory concentration (MIC), i.e., the lowest concentration of an antimicrobial that will inhibit microbial growth after overnight incubation, of all samples is shown in Table 2. All of the oil samples exhibited satisfying antimicrobial activity due to low MICs (<12.5% of initial concentration) against sensitive microorganisms. Furthermore, both of the oil samples have multiple lower MIC values for the same sensitive microorganism compared with those of the extracts and HYs. The only exception was *S. montana* subsp. *montana* SWE, where the MIC value for *B. cereus* ATCC 11778 was eight times lower compared with the corresponding oil sample. According to the results in Table 2, it can be concluded that the stronger antimicrobial activity has *S. montana* subsp. *variegata*.

Based on the obtained MIC value of the different samples, the antimicrobial activity was additionally tested in view of the pharmacodynamic potential of antimicrobial activity using a time–kill kinetics study (Appendix A, Figure 4 and Figure 5). Kinetic modeling was performed for a time-relative response of microbial growth inhibition during 24 h for bacteria and 72 h for yeast.

According to the results present in Figure 4 and Figure 5 and Table 3, all sensitive bacteria at MIC of *S. montana* subsp. *montana* showed a reduction in the number of viable cells over the first 12 h. On the other hand, contact time for the same effect with subsp. *variegata* was lower for all bacteria. Briefly, the suppression of *B. cereus*, *S. aureus*, *E. faecalis*, and *E. coli* required a contact time of 6 h, while *S*. Typhimurium needed only 4 h.

Similar pathways were observed for yeasts, where 18 or 12 h was necessary for killing all of the viable cells in contact with *S. montana* subsp. *montana* and subsp. *variegata*, respectively. The pharmacodynamic profile of HYs of both tested *S. montana* against *S. cerevisiae* and *C. albicans* at their MIC values followed similar pathways with complete reduction within 48h. On the contrary, the behavior of *B. cereus* viability during contact with SWE extracts was different, indicating a reduction of cells within 12 or 24 h, depending on the subspecies. Furthermore, a similar difference was observed during contact between the UEA-MeOH extract of subsp. *variegata* and the two tested yeast strains. Briefly, a complete reduction in viability of *S. cerevisiae* was noticed within 18 h, while the same effect for *C. albicans* was gained for 48 h.

For a better understanding of the kinetics pathways of pharmacodynamics potential of tested *S. montana*, mathematical modeling was undertaken, while the gained models can be described as accurate. This result is evident in Table 3, which summarizes the regression coefficients for the obtained models clarifying the speed and intensity of the MICs effect during contact time.

Additionally, the quality of the mathematical models was determined using several parameters, which described the fit between the experimental results and model gained results (Appendix A). Overall, the models shown in Figure 3 and Figure 4 fitted well with the experimental data and, consequently, adequately predicted the targeted outputs. Additional confirmation of these results was the fact that for the proposed models of the pharmacodynamical potential modeling, the minimum value of the coefficient of determination was 0.94.

## 3. Discussion

*S. montana* essential oil exhibits large variations in the relative concentrations of its principal components [16]. Variability in chemical composition is evident in *S. montana* subsp. *montana* and subsp. *variegata* collected in Croatia and Italy [8,12]. Differences are notable even within a single population found in some areas because of expressed polymorphism, as well as in populations coming from distant habitats [12,16]. Further, the weather conditions during the year can influence the chemical composition of the essential oils [47,48,49].

*Satureja* species are a very important source of essential oils and other biologically active molecules [50]. The occurrence of a significant number of phytochemicals in plants can have a positive impact on human health if they are part of a daily diet manifesting their antioxidant activity [51]. Using a simple and easy-to-do method, the total phenolic content of all of the samples was determined (Figure 1). The results from this study are comparable with the previously gained results, where 220.95 mg GAE/g was obtained [37].

Since the extraction methods such as SWE, UAE, and MAE are relatively recent and insufficiently applied, there are no previous comparative reports on the bioactivity of the *S. montana* subsp. *Montana* and *S. montana* subsp. *ariegate* extracts. Almost all of the extracted samples of *S. montana* subsp. *montana* had a higher phenolic content compared with the extracts of *S. montana* subsp. *variegata* (Figure 2). The inverse results were obtained for the phenolic content in the EO samples; *S. montana* subsp. *variegata* had significantly high phenolic content compared with that of *S. montana* subsp. *montana*. In general, it can be observed that the most effective technique to produce a high phenolic extract of *S. montana* plants was subcritical water extraction. The major advantage of the SWE technique is the use of a solvent that is non-toxic, and therefore, it is more suitable for extraction. Furthermore, the use of SWE extraction represents a better option in view of the environment, economy, and safety [52].

The phenolic yield in the MAE samples was at the same level, indicating the independence of this extraction technique from the solvent. On the other hand, a significant difference was observed with the UAE samples, where using methanol as solvent had better extraction efficiency compared to water with respect to the total phenolic content. This indicates that the solvent had a strong influence on the UAE extraction, which is rather unexpected, considering that a solvent is mainly a medium that transmits sound waves. This entails the assumption that the solubility of total phenols in 70% methanol as solvent of these bioactives is more dominant for extraction efficiency compared to the extraction techniques. Slightly higher TPC was observed in both methanolic extracts resulting from the two different extraction methods, UAE and MAE. Using conventional extraction with 70% methanolic extract of *Satureja* biomass, the average value of the total phenolic content was about 25 mg GAE/g [53], indicating that using alternative extraction techniques coupled with methanol as solvent could improve the extraction of the phenolic compounds. A better extraction rate of phenols with methanol was also observed in the case of conventional extraction compared with extraction with chloroform [28]. In summary, we observed a significant impact of the extraction methods on phenolic content, which contradicts previous studies. This also can indicate differences in extraction techniques and solvent levels, but also most probably variances in used standards, particle size distribution in samples, herbal parts, conditions, etc.

Plant extracts are progressively becoming vital additives in pharmaceutical and food industries since they contain polyphenols that have antimicrobial and antioxidant activity [54]. In general, the biological activity of local and worldwide spread plant varieties, especially those used as traditional medicinal plants, is considered presently imperative for drug and food supplement development [55]. Additionally, screening antioxidant activity based on different assays such as DPPH, ABTS, reducing power according, and superoxide anion offer the possibility to determine the frame in the design of research on bioactive compounds, especially assessing the effect of extract obtaining on the stability of phenolic compounds [56]. Taking into account the need to estimate the antioxidant capacity of an extract as comprehensively as possible, the four mentioned antioxidant assays were conducted, and the gained results are presented in Figure 2. The methods used offer the chance to spot different mechanisms, kinetics, and antioxidant molecules with different polarities [57].

Plant species that are rich in phytochemicals have natural mechanisms to protect themselves against various pathogenic microbes by their identified antimicrobial activity. Based on the results presented in Figure 2, the ABTS assay showed higher activity than those for the DPPH assay regarding the identical principles that these two assays represent. A study using *S. kitaibelii* showed that the medium for the assay had a crucial impact on the solubility and availability of the extracted compounds [58]. The reducing power varied from 0.1 μmol TE/g for HY to 369.31 μmol TE/g for SWE of *S. montana* subsp. *montana*. The minimal and maximal RP values for subsp. *variegata* were 0.144 and 397.68 μmol TE/g for HY and UAE-H_2_O, respectively. The most significant variability was present between subspecies for the UAE-H_2_O and UAE-MeOH extraction methods. The DPPH assay showed small differences between investigated subspecies; only samples obtained with UAE-H_2_O had different values, whereas *S. montana* subsp. *montana* has 61.18 μmol TE/g while *S. montana* subsp. *variegata* had 17.14 μmol TE/g. The highest values for *S. montana* subsp. *montana* was obtained from EO, while the highest values from *S. montana* subsp. *variegata* was gained within UAE-H_2_O, but its EO also shows high activity. The results of the SOA assay exhibited similar characteristics to the DPPH assay, where the samples obtained with UAE-H_2_O differed, with values of 226.66 μmol TE/g and 127.19 μmol TE/g for *S. montana* subsp. *montana* and *S. montana* subsp. *variegata*, respectively. For both subspecies, the highest results of the SOA assay were obtained by the SWE extraction technique.

Comparing the obtained results with the previously reported in the scientific literature, some correlation can be seen. In brief, the reducing power of a methanol extract of *Satureja montana* was recorded to be 7.089 mg/mL [37]. The DPPH values for EOs of *S. montana* subsp. *montana* and subsp. *variegata* from Italy were 62.02 and 53.01 μmol TE/g, respectively [8]. The results of the ABTS test for EOs show 107.91 and 92.3 μmol TE/g [8], while in this study, there are 2446.23 and 2587.47 μmol TE/g for *S. montana* subsp. *montana* and subsp. *variegata*, respectively. In addition, a better effect has been noticed in the ABTS than in the DPPH test for both EOs [59].

As a good source of phenols, which can be attributed to this inhibitory activity against microorganisms, the in vitro antimicrobial activity of the *S. montana* samples was qualified by inhibition zone diameters, the minimal inhibitory concentration (MIC), and the pharmacodynamics pathways of biocide effect (Figure 3, Figure 4 and Figure 5, Table 2 and Table 3, Appendix A). Differences in the chemical compositions reflect the variability within the genus *Satureja*, but also the variability within the same species, which was correlated with the results of the antimicrobial activity of the tested samples in this study. In view of the antimicrobial testing, this observation of the sensitivity of Gram-positive bacteria to *S. montana* was in agreement with the published research [28,29]. The main reason for the lower susceptibility of Gram-negative bacteria, such as *Pseudomonas aeruginosa*, can be explained by cell membranes rich in hydrophilic lipopolysaccharides. This structure represents a barrier between macromolecules and hydrophobic compounds. In contrast to this, Gram-positive bacteria have greater permeability of the cell wall since the main structural units are peptidoglycans, which are covalently linked to teichoic and teichuronic acids [29].

The EOs of the two *S. montana* subspecies had relatively high antimicrobial activities against almost all tested bacterial strains and on two yeast species. In the case of *S. montana* subsp. *montana* oil, similar antimicrobial behavior against *Bacillus cereus* and *Staphylococcus aureus* was reported [18,28,29,30,31,33,34,35]. Additionally, the antimicrobial activity against *Escherichia coli* was not uniform. Briefly, some researchers reported high antimicrobial activity of the mentioned oil against *E. coli* [29,30,31,33,34,35], but some others did not find such results [18,28].

*E. faecalis* showed sensitivity to *S. montana* essential oil [30,31], which is in agreement with the results of this study (Figure 3). Regarding present knowledge, only one study has tested *Salmonella* Typhimurium [36], and obtained results that are in agreement with this study, i.e., it could be concluded that *S. montana* essential oil demonstrated high sensitivity to this pathogenic bacterium. The only exception is the lower antibacterial activity against *P. aeruginosa*, which was also resistant to the tested antibiotic combination. The minimal or average antibacterial activity was reported against *P. aeruginosa* reference strains [30,31,33,35,37]. The equal activity against *Saccharomyces cerevisiae* and *Candida albicans* has been noted in several studies [30,31,32]. Furthermore, no antifungal activity of both tested EOs was reported. The potential reason for fungal resistance on the tested EOs can be explained by limiting the interactions between the fungal structures and the low concentrations of active substances [60]. However, the fungistatic to fungicide effect EOs of *S. montana* subsp. *montana* using oil concentration in the range of 150 to 300 μg/mL against nine phytopathogenic fungi has also been reported [38].

Serrano et al. [37] reported that aqueous and methanolic extracts did not have antibacterial activity. Silva et al. [61] also tested the HY of *S*. *montana* subsp. *montana*, as a by-product of EO production, reported minimal antibacterial activity against *B. cereus*, *E. faecalis*, *E. coli*, *L. monocytogenes*, and *P. aeruginosa* (inhibition zone between 10–12 mm).

The concept of determining the pharmacodynamics potential of some antimicrobial compounds is based on monitoring the time–kill kinetic study as one of the main parameters for antimicrobial pharmacodynamics is the obtained MIC value of the targeted microorganism. It is believed that the plant concentration at the site of the microbial activity must be at MIC level or higher to obtain an antimicrobial effect [62]. However, each microorganism relates to the inhibition effect in a different way, which is the base for pharmacodynamics or time–kill kinetics studies. For example, the total biocide effect can be observed during the initial contact between the plant sample and microbial cell, but also after a certain period of time. Furthermore, some antimicrobials have prolonged persistent effects in which the inhibition of growth continues after the plant concentration level falls below the MIC [63]. Briefly, the establishment of the rate at which a microorganism is killed by a test substance as a function of survival data recorded at enough contact time points such that a graph can be constructed modeling the decline in population over time to the point of extinction [64]. As the following step in the pharmacodynamic study can be directed to investigate using the lower concentration that MICs (in order to check the possibility for biostatic effect during time and cost reduction) or higher concentration of the obtained MICs (in order to obtain the more effective biocide effect).

In summary, the results in this study imply that the strong antimicrobial activity of the EO and SWE samples can determine the further use of both *S. montana* subspecies as antimicrobial agents in food or/and pharmaceutical industries. On the other hand, the natural antimicrobials in these samples indicated the possibility of the study for the potential inhibition of phytopathogens microorganisms causing different plant diseases.

## 4. Materials and Methods

### 4.1. Plant Material

The two most widely cultivated subspecies of *Satureja* in Serbia were grown in the experimental fields of the Institute of Field and Vegetable Crops Novi Sad (IFVCNS), located in Bački Petrovac (Vojvodina Province, North Serbia). *Satureja montana* subsp. *montana* (Vouch No 2-1561) and *S. montana* subsp. *variegata* (Host.) P.W. Ball (Vouch No 2-1388) were authenticated by Milica Rat and deposited at the botanical collection of the Herbarium BUNS (University of Novi Sad).

The seeds (IFVCNS collection) were sown in the greenhouse in February 2018, and the seedlings were transplanted in a field at the end of April. During the vegetation season in the first year of cultivation, only irrigation and manual weed controls were applied. In the second year of cultivation, the hoeing and weeding were performed 3 times before the plants were in the full flowering stage. *Satureja montana* subsp. *montana* reached the flowering stage first, and after approximately two weeks, *S. montana* subsp. *variegata* bloomed. The harvest was performed manually in the full flowering stage by cutting plants 5–10 cm above the ground. After harvest, the plants were transported to a solar dryer (temperature up to 40 °C). After two days, the dried plant material was stored in multilayer paper bags at room temperature until further processing.

### 4.2. Extraction

#### 4.2.1. Steam Distillation

Dry aboveground flowering plant material of *S. montana* was processed by steam distillation, using the procedure described by Aćimović et al. [65]. In brief, 100 kg of the plant material was used in the process that lasted 3 h. The obtained EOs were yellow in color for both subspecies, with a characteristic *strong* fragrance. It was decanted and filtrated over anhydrous sodium sulphate and stored in dark glass vials at 4 °C until further analysis. HYs, which occur as by-products during the steam distillation process, were collected in a Florentine flask together with EOs. After decanting the EO, the HY, which possesses a fragrance similar to the corresponding oil, was stored in sterile plastic containers at 4 °C until further analysis.

#### 4.2.2. Subcritical Water Extraction (SWE)

The subcritical water extraction of *S. montana* was performed in a home-built subcritical water extractor as described previously [58], maintaining a sample-to-solvent ratio of 1:30 in all extractions. The pressurization of the extraction vessel to 20 bars was performed with 99.999% nitrogen (Messer, Germany). The extraction vessel was filled with the plant sample and water and was closed tightly, after which it was pressurized with nitrogen through the built-in valve. After pressurization, a working temperature (130 °C) was set to temperature programmer. After reaching a working temperature, extraction lasted for 30 min. The vessel was heated at approximately 10 °C/min increments. Agitation was assured at the frequency of the vibrational platform of 3 Hz. At the end of the extraction, the process vessel was immediately cooled in a flow-through water bath at 20 ± 2 °C. The depressurization was performed by valve opening and purging nitrogen through a valve. The obtained extracts were separated by filtration through Whatman qualitative filter paper, grade 1, and stored in a refrigerator at 4 °C for further analysis.

#### 4.2.3. Ultrasound-Assisted Extraction (UAE)

The sample (5 g) was placed into a round bottom flask, and the solvent was added, maintaining the sample-to-solvent ratio to 1:30, as in all other extractions. The extractions were performed with double distilled water and with 70% aqueous methanol (UAE-MeOH and UAE-H_2_O, respectively). The mixture in a flask was placed into an ultrasound water bath (Iskra, Slovenia) at the position nearest possible to the ultrasound generator, connecting a condenser. The flask was fixed in position, and the extraction proceeded for 30 min at room temperature (constant temperature, frequency of 40 kHz, ultrasound power of 90% (216 W). The obtained extracts were separated by filtration through Whatman qualitative filter paper, grade 1, and stored in a refrigerator at 4 °C for further analysis.

#### 4.2.4. Microwave-Assisted Extraction (MAE)

The microwave-assisted extraction was performed in a modified microwave system (LG Electronics), placing the sample/solvent mixture in a flask and connecting it to a condenser. For comparison purposes, all of the parameters for different extraction systems were maintained the same, such as the sample-to-solvent ratio and the extraction time. In MAE, both double distilled water (for comparison with SWE) and 70% aqueous methanol (Merck) were used to clearly distinguish between the influence of both the solvent type and the extraction technique (MAE-MeOH and MAE-H_2_O, respectively). Prior to extraction, the flow of cold tap water through the condenser was assured to prevent solvent evaporation and losses of volatiles. The extraction was performed for 30 min at 450 W magnetron power. The obtained extracts were filtrated through Whatman qualitative filter paper, grade 1, and stored in a refrigerator at 4 °C for further analysis.

### 4.3. Chemical Analysis

The volatile fraction of the EO and HY were analyzed by GC/FID and GC-MS, as previously described by Aćimović et al. [65,66,67,68]. The non-volatile compounds were analyzed in terms of total phenolic content by the spectrophotometrically microscale-adapted Folin–Ciocalteau method [69]. The results of the total phenolic content in the extracts (SWE, UAE, MAE) were expressed as gallic acid equivalents (GAE) per g of plant sample. Before analyzing, the essential oils were dissolved in methanol at the concentration of 250 mg/mL, and the results were expressed as GAE per g of essential oil, while for hydrolates the results were expressed as GAE) per mL.

### 4.4. In Vitro Examination of Antioxidant Activity

For the determination of the antioxidants, the samples were analyzed using a few different methods. Briefly, the antioxidant capacity was analyzed using 2,2-diphenyl-1-picrylhydrazyl (DPPH) and 2,2-azino-bis-3-ethylbenzothiazoline-6 acid (ABTS) [70], reducing power [71], and superoxide anion method [72]. The tests were performed with SWE, UAE, MAE extracts, essential oils dissolved in methanol at the concentration of 250 mg/mL, and hydrolates. For all of the antioxidant assays, the results were expressed in μmol Trolox equivalent (TE) per g of plant sample or per g of essential oil, while for hydrolates the results were expressed as GAE) per mL.

### 4.5. In Vitro Examination of Antimicrobial Activity

#### 4.5.1. Test Microorganisms

The antimicrobial activity of the different extracts of *S. montana* was determined against a wide specter of microorganisms: *Escherichia coli* ATCC 25922, *Pseudomonas aeruginosa* ATCC 27853, *Salmonella* Typhimurium ATCC 13311, *Bacillus cereus* ATCC 11778, *Staphylococcus aureus* ATCC 25923, *Enterococcus faecalis* ATCC 19433, *Saccharomyces cerevisiae* ATCC 9763, *Candida albicans* ATCC 10231, as well as *Aspergillus brasiliensis* ATCC 16404. All of the cultures were stored at a temperature of −80 °C using glycerol as a cryoprotectant. Prior to microbiological analysis, all of the bacterial strains were grown on Müller–Hinton Agar (HiMedia, Mumbai, India) at 37 °C for 24 h, except *B. cereus*, which was incubated at 30 °C for 18 h. For the chosen eukaryotes, Sabouraud Maltose Agar (HiMedia, Mumbai, India) was used, and the strains of yeasts and fungi were incubated at 30 °C for 24 h and 25 °C for 5 days, respectively.

#### 4.5.2. Antimicrobial Screening

As a preliminary step in a screening of the antimicrobial potential of different *S. montana* extracts, the disc-diffusion method was chosen and completely performed as described by Micić et al. [73]. The most important parts of the described method imply the usage of the freshly prepared suspensions of overnight cultures for the inoculation of adequate nutrient medium in a Petri dish and the application of the tested extracts (15 μL) onto three sterile discs for each microorganism previously applied on the medium. After the period of incubation, halo zones around the discs were measured, and the obtained results of the antimicrobial activity were interpreted as follows: resistant-inhibition zone lower than 22 mm, intermediary effect-inhibition zone between 22 and 26 mm, and sensitive-inhibition zone greater than 26 mm.

#### 4.5.3. Minimal Inhibitory Concentrations

The minimal inhibitory concentration was determined for the microorganisms that were sensitive to the initial concentration of the extract. Different decreasing concentrations of the extract were tested (100–0.78%) using the microdilution method, which was previously described by Micić et al. [73]. The MIC value was determined using Equation (1), involving a number of microorganisms in the positive control for microbial growth (*N_c_*) and treated sample (*N_t_*).
(1)$$MIC=\frac{N_{c}-N_{t}}{N_{c}}\cdot 100\;(\%)$$

#### 4.5.4. Pharmacodynamics Potential of Antimicrobials

The procedure for the determination of the pharmacodynamic potential of a substance with an antimicrobial effect was performed for the previously obtained MIC value [47]. The contact time between the extracts and microorganisms was conducted at 37 °C for 24 h (bacteria) or 30 °C for 72 h (yeast and fungi). For the sampling times, the values were used as recommended by Ferro et al. [74]. For a better understanding of microbial behavior during contact with antimicrobials, mathematical modeling has been applied. Briefly, the four-parameter sigmoidal numerical model represents a suitable tool for the determination of the behavior of biological systems and described in detail by Romano et al. [75] and can be summarized in the form of Equation (2), where *y*(*t*) presents the bacterial and yeast growth in the presence of MICs of antimicrobials (markers signify the experimental data; lines indicate predictive results), during the process.
(2)$$y(t)=d+\frac{a-d}{1+{\left(\frac{t}{c}\right)}^{b}}$$

The four-parameter logistic regression function is suitable for biological systems, which fits data to the S-shaped curves. The regression coefficients which participate in the function could be described as follows: *a*—the minimum value that was obtained (close to *t* = 0), *d*—the maximum value that could be obtained (t=∞), *c*—the inflection point (the point on the S-shaped curve between *a* and *d*) and *b* = the Hill’s slope of the curve (the steepness of the curve at point *c*).

## Figures and Tables

**Figure 1 antibiotics-11-01235-f001:**
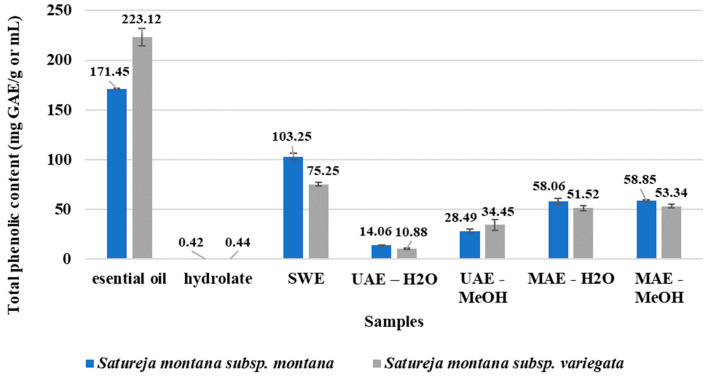
Total phenolic content.

**Figure 2 antibiotics-11-01235-f002:**
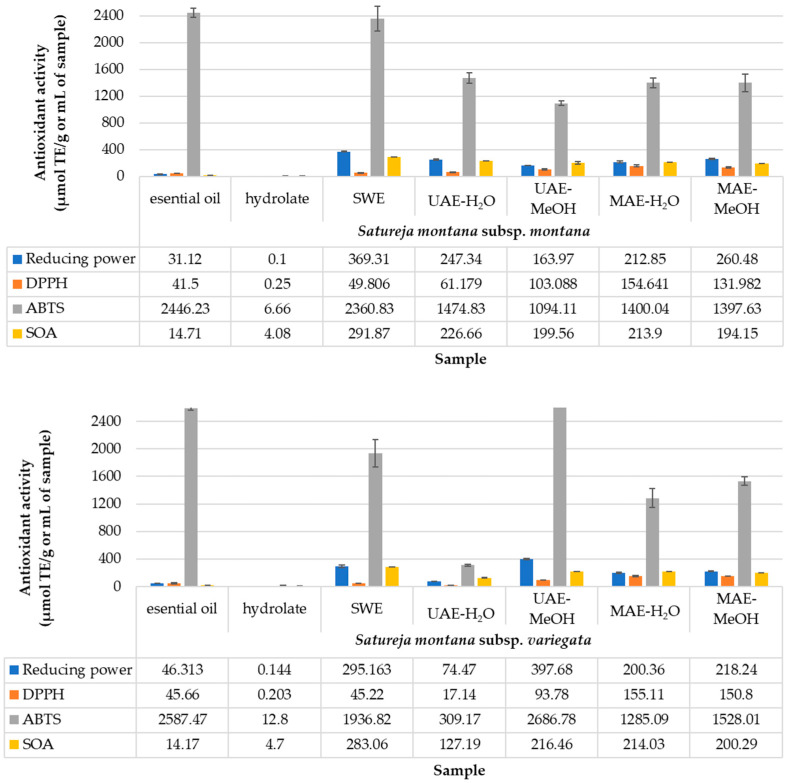
Antioxidant activity.

**Figure 3 antibiotics-11-01235-f003:**
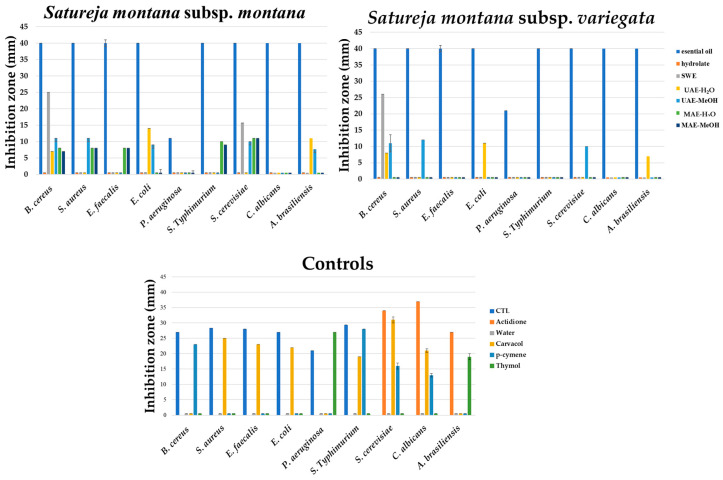
In vitro screening of antimicrobial activity of *S. montana* samples.

**Figure 4 antibiotics-11-01235-f004:**
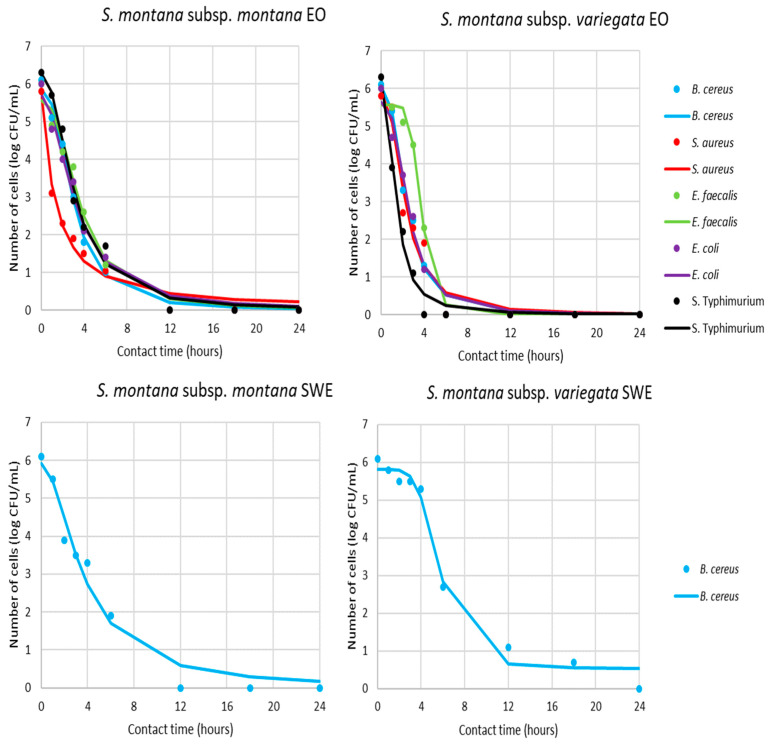
Time–kill kinetics of bacterial growth in presence of MICs of antimicrobials (markers signify the experimental data; lines indicate predictive results).

**Figure 5 antibiotics-11-01235-f005:**
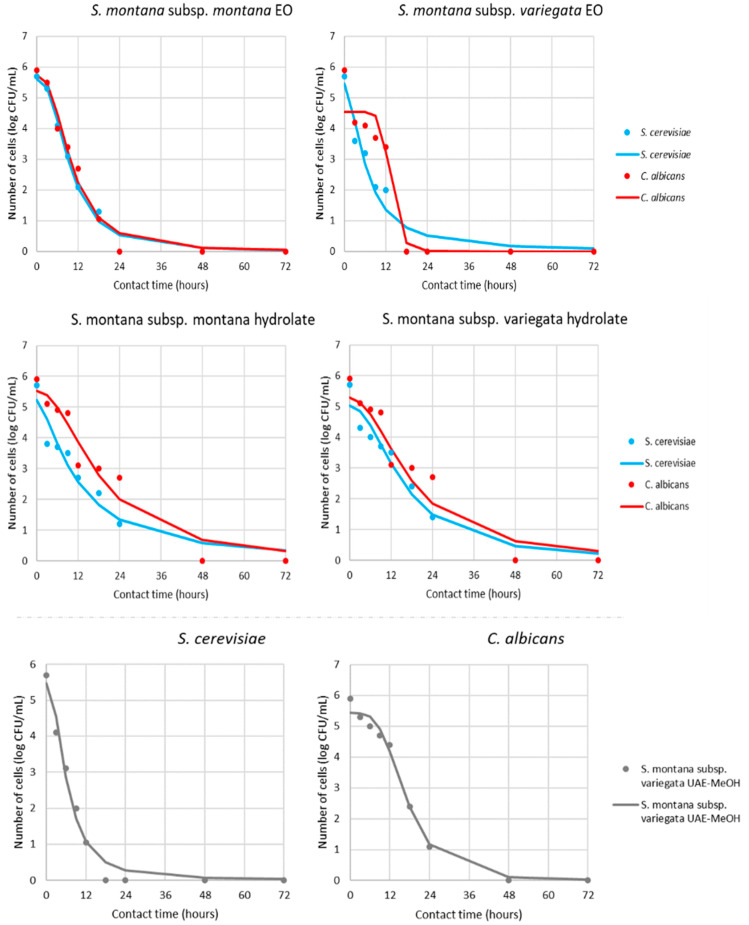
Time–kill kinetics of yeast growth in presence of MICs of antimicrobials (markers signify the experimental data; lines indicate predictive results).

**Table 1 antibiotics-11-01235-t001:** Volatile compounds from *S. montana* essential oil (EO) and hydrolate (HY).

№	Compound	RI	Subsp. *montana*		Subsp. *variegata*	
			EO	HY	EO	HY
1	2-methyl-Butanoic acid	839	-	0.1	-	-
2	α-Thujene	924	0.3	-	1.1	-
3	α-Pinene	931	0.8	-	0.7	-
4	Camphene	945	0.4	-	0.4	-
5	1-Octen-3-ol	974	1.6	1.3	1.0	0.7
6	Myrcene	988	0.9		1.4	-
7	3-Octanol	993	0.1	0.1	-	0.1
8	α-Phellandrene	1003	0.1	-	0.2	-
9	δ-3-Carene	1009	0.1	-	0.1	-
10	α-Terpinene	1014	1.4	-	1.7	-
11	p-Cymene	1023	32.3	0.5	10.7	0.3
12	Limonene	1026	1.0	-	0.7	-
13	1,8-Cineole	1028	0.7	0.1	0.2	tr
14	cis- β -Ocimene	1034	-	-	0.1	-
15	γ-Terpinene	1055	10.5	-	7.3	-
16	cis-Sabinene hydrate (IPP vs. OH)	1063	0.1	-	0.4	-
17	cis-Linalool oxide (furanoid)	1070	-	0.1	-	-
18	trans-Linalool oxide (furanoid)	1087	-	0.1	-	-
19	Terpinolene	1086	0.1	-	0.1	-
20	p-Cymenene	1087	0.1	-	-	-
21	Linalool	1097	2.2	0.6	0.3	0.2
22	Camphor	1142	0.1	tr	0.1	-
23	Menthone	1151	0.1	-	-	-
24	Borneol	1163	2.1	1.0	1.1	0.3
25	Menthol	1169	0.5	tr	-	-
26	Terpinen-4-ol	1174	1.1	1.2	0.6	0.9
27	α-Terpineol	1188	0.2	0.1	0.2	0.2
28	Thymol, methyl ether	1233	-	-	0.1	-
29	Cumin aldehyde	1239	0.1	-	-	-
30	Carvacrol, methyl ether	1242	0.3	-	0.3	-
31	Thymol	1292	0.2	2.1	51.4	0.5
32	Menthyl acetate	1296	0.2	-	-	-
33	Carvacrol	1304	35.7	91.6	13.1	96.4
34	Thymol acetate	1352	-	-	0.1	-
35	α-Ylangene	1369	0.1	-	-	-
36	α-Copaene	1374	0.2	-	-	-
37	β-Bourbonene	1382	0.1	-	tr	-
38	trans-Caryophyllene	1417	2.1	-	2.1	-
39	β-Copaene	1426	0.1	-	-	-
40	α-trans-Bergamotene	1434	0.1	-	-	-
41	Aromadendrene	1437	0.1	-	-	-
42	α-Humulene	1452	0.1	-	0.1	-
43	γ-Muurolene	1475	0.3	-	0.1	-
44	β-Selinene	1485	0.1	-	-	-
45	Viridiflorene	1494	0.2	-	-	-
46	α-Muurolene	1499	0.1	-	-	-
47	β-Bisabolene	1507	0.9	-	3.1	-
48	γ-Cadinene	1513	0.2	-	0.1	-
49	δ-Cadinene	1522	0.5	-	0.1	-
50	α-Calacorene	1542	0.1	-	-	-
51	Spathulenol	1575	tr *	-	-	-
52	Caryophyllene oxide	1581	0.5	-	0.2	-

EO—essential oil; HY—hydrolate; * tr—trace (less than 0.05%).

**Table 2 antibiotics-11-01235-t002:** Minimal inhibitory concentration (%) *.

*S. montana*		Microorganisms						
		Gram-Positive Bacteria			Gram-Negative Bacteria		Yeasts	
Sample type	Subsp.	*B. cereus* ATCC 11778	*S. aureus* ATCC 25923	*E. faecalis* ATCC 19433	*E. coli* ATCC 25922	*S.* Typhimurium ATCC 13311	*S. cerevisiae* ATCC 9763	*C. albicans* ATCC 10231
Essential oil	*montana*	12.5	6.75	6.75	12.5	12.5	1.56	3.375
	*variegata*	1.56	1.56	1.56	6.75	6.75	0.78	0.78
Hydrolate	*montana*	>100 *					25	25
	*variegata*						12.5	12.5
SWE	*montana*	1.56	>100					
	*variegata*	6.75						
UAE-MeOH	*montana*	>100						
	*variegata*	>100					50	50
UAE–H_2_O	*montana*	>100						
	*variegata*							
MAE–MeOH	*montana*							
	*variegata*							
MAE–H_2_O	*montana*							
	*variegata*							

* The initial concentration was determined as 100%.

**Table 3 antibiotics-11-01235-t003:** The regression coefficients for kinetics modeling for pharmacodynamical potential of *S. montana* samples.

Sample	Microorganism		Regression Coefficient			
		Subsp.	d	a	c	b
	*B. cereus*	*montana*	0.00	5.864	2.986	2.408
		*variegata*	0.00	6.057	2.314	2.503
	*S. aureus*	*montana*	0.00	5.751	1.348	1.128
		*variegata*	0.00	5.943	2.238	2.240
Essential oil	*E. faecalis*	*montana*	0.00	5.559	3.683	2.386
		*variegata*	0.00	5.573	3.753	6.435
	*E. coli*	*montana*	0.00	5.724	3.182	1.965
		*variegata*	0.00	5.634	2.540	2.658
	*S.* Typhimurium	*montana*	0.00	6.293	3.087	2.148
		*variegata*	0.00	6.247	1.344	2.174
	*S. cerevisiae*	*montana*	0.00	5.627	9.551	2.452
		*variegata*	0.00	5.454	6.268	1.699
	*C. albicans*	*montana*	0.00	5.745	10.017	2.472
		*variegata*	0.00	4.540	13.253	9.102
Hydrolate	*S. cerevisiae*	*montana*	0.00	5.221	11.737	1.468
		*variegata*	0.00	5.026	15.569	2.012
	*C. albicans*	*montana*	0.00	5.530	18.165	2.024
		*variegata*	0.00	5.280	17.690	2.011
SWE	*B. cereus*	*montana*	0.00	5.929	3.680	1.863
		*variegata*	0.537	5.818	5.694	5.116
UAE-MeOH	*S. cerevisiae*	*variegata*	0.00	5.479	6.244	2.166
	*C. albicans*	*variegata*	0.00	5.432	16.807	3.617

## Data Availability

Not applicable.

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
