# Peer review of "Variability in Biological Activities of Satureja montana Subsp. montana and Subsp. variegata Based on Different Extraction Methods"

_antibiotics, 2022, doi:10.3390/antibiotics11091235_

Round 1

Reviewer 1 Report

The manuscript entitled Variability in biological activities of Satureja montana subsp. montana and subsp. variegata based on different extraction methods brings an interesting perspective regarding the potential use of several herbal preparations obtained from S. montana as antibacterial agents whit promising applications in foods or pharmaceutical industry and also highlights the impact of processing methods on the final quality of the final products in terms of bioactive potential. However, my opinion is that this work could be improved taking into account the following suggestions:

L20: were used

L353: were determined / were authenticated;

L368: Is that correct the amount of plant material? It is 1.00 kg or 100 (one hundred) kg?

L372: were collected

L370-374: Why is this part written as italic? Maybe might be corrected?

Introduction

-        Taking into account the research topic, I recommend to the authors to make a comprenhensive description of the state-of-the-art regarding the antimicrobial potential of S. montana and emphasize some aspects linked to the focus of their research (i.e. the impact of the extraction method or the herbal preparation on the antibacterial potency, the benefits and the potential risks of the use of these species as antibacterial agent – if they exist).

Results:

-        Section 2.1.2. The section needs to be properly named. It is supposed that you are measuring the non-volatile phenolic content, but at the same time you have made the quantitative determination of TPC from a volatle oil and a hydrolate. How you can make a difference between the phenolics from volatile and non-volatile fractions when you measure them using Folin-Ciocîlteu method? There is a lack of concordance between the title of the section and the content, please revise it.

-        Moreover, the results are expressed as mg GAE/g or mL”... what this means? As I can see, all the extacts were tested as liqiud preparations, so, I don’t understand which of them where wheighed and which of them were measures in terms of volumes. Please, check again the units and try to explain which unit was used for which extract.

-        Section 2.2: Same problem here with the way that you extressed the results... it must be clearly made a difference between volume and mass and which extracts were tested in terms of mass and volume. Also, the standard compound (Trolox) is not mentioned neither in „Materials and method” and „Results” section... please, revise it.

Discussion

-        The relevance and significance of the pharmacodynamic model applied in this research must be more described. It was previoully applied to evaluate the antimicrobial potential of the herbal preparations obtained from this species or other related species? Which are the strong points and the limitations of this model when it is used evaluate the antimicrobial potential of phytochemicals or phytopreparations?

Materials and methods:

-        Section 4.2.3: the extraction parameters need to be more described; can the authors mention the amplitude or/and the frequency used for develop UAE? Also, during UAE, the temperature was monitored, it was constant? It is well known that ultrasound treatment induces a thermal effect in the extraction mixture that influences the extraction yield and the final quality of the extract so, how it was monitored and quantified this effect?

-        Section 4.3. Could the authors bring a more appropriate chemical description for the SWE, UAE-MeOH, UAE-H2O, MAE-MeOH and MAE-H2O? As long as for the volatile compounds they provided an in-depth characterization of individual chemical components of volatile oils, for the uniformity of research design it could be added an more detailed description of phenolic profile for the afforementioned extracts.

-        Section 4.3: The analysis of TPC is not clearly described; the cited method is used for the assessment of aqueous solutions, hence, how the authors analyzed the volatile oils as long as this herbal preparations are not miscible with aqueous reagents used in this assay?  Also, it must be mentioned the standard compound used to express the results (I guess it is gallic acid based on the section ”Results”) This section must be improved.

-        Section 4.4: Same as previous, all these methods involve testing the antioxidant potential in aqueous solution. Please, provide a more detailed description of the methodology used in this section.

-        Section 4.5.4: Please, define the coeficients from the equation. Also, as I can see in the cited reference, the model is developed for describing the kinetics of an leavening process so, how this can be extrapolated to evaluate the pharmacodynamics of an antibacterial agent? Please, make an in-depth description of this method.

Author Response

Reviewer 1

The manuscript entitled Variability in biological activities of Satureja montana subsp. montana and subsp. variegata based on different extraction methods brings an interesting perspective regarding the potential use of several herbal preparations obtained from S. montana as antibacterial agents whit promising applications in foods or pharmaceutical industry and also highlights the impact of processing methods on the final quality of the final products in terms of bioactive potential.

The Authors would like to thank the Reviewer for a quick and professional review as well as the opportunity to make essential and crucial changes in our work. All the Reviewer' remarks are accepted and the paper is changed according to their comments. The Authors believe that the changed paper would satisfy the Reviewer' criteria and that it is going to be interesting enough for publishing in the Processes.

We decided to revise the manuscript according to the Reviewer' remarks, highlighting the changes directly in the revised manuscript.

However, my opinion is that this work could be improved taking into account the following suggestions:

L20: were used

ANSWER:  Corrected.

L353: were determined / were authenticated;

ANSWER: Corrected.

L368: Is that correct the amount of plant material? It is 1.00 kg or 100 (one hundred) kg?

ANSWER:  We used 100 (one hundred) kg of plant materias due to the fact that steam distillation was done as semi-industrial process.

L372: were collected

ANSWER: Corrected.

L370-374: Why is this part written as italic? Maybe might be corrected?

ANSWER: It is a mistake, we corrected, thank you for this observation.

Introduction

-        Taking into account the research topic, I recommend to the authors to make a comprenhensive description of the state-of-the-art regarding the antimicrobial potential of S. montana and emphasize some aspects linked to the focus of their research (i.e. the impact of the extraction method or the herbal preparation on the antibacterial potency, the benefits and the potential risks of the use of these species as antibacterial agent – if they exist).

ANSWER: Thank you for this suggestion, we added a paragraph about the antimicrobial activity of Satureja montana. Neverless, the detailed explanation of antimicrobial effect of different samples type is involved in disscusion part where comparation of the obtained results and scientific literature was shown.

Results:

-        Section 2.1.2. The section needs to be properly named. It is supposed that you are measuring the non-volatile phenolic content, but at the same time you have made the quantitative determination of TPC from a volatle oil and a hydrolate. How you can make a difference between the phenolics from volatile and non-volatile fractions when you measure them using Folin-Ciocîlteu method? There is a lack of concordance between the title of the section and the content, please revise it.

ANSWER: Thank you for this observation. We measured the total phenolic content in all samples, so we corrected the name of the subsection.

-        Moreover, the results are expressed as “mg GAE/g or mL”... what this means? As I can see, all the extacts were tested as liqiud preparations, so, I don’t understand which of them where wheighed and which of them were measures in terms of volumes. Please, check again the units and try to explain which unit was used for which extract.

ANSWER: Thank you for your comment. We added an explanation in the Material and methods section:

The results of total phenolic content in extracts were expressed as gallic acid equivalents (GAE) per g of plant sample. Before analyzing, the essential oils were dissolved in methanol at the concentration of 250 mg/mL, and the results were expressed as GAE per g of essential oil, while for hydrolates the results were expressed as GAE) per mL.

The extracts were tested as liquid preparations, but we calculated the total phenolic content per dry plant material. For essential oil, we expressed results also in grams (according to its density), while for hydrolates we used mL, since we do not know the concentration.

-        Section 2.2: Same problem here with the way that you extressed the results... it must be clearly made a difference between volume and mass and which extracts were tested in terms of mass and volume. Also, the standard compound (Trolox) is not mentioned neither in „Materials and method” and „Results” section... please, revise it.

ANSWER: Thank you! We added the proposed information:

      For all antioxidant assays, the results were expressed in μmol Trolox equivalent (TE) ) per g of plant sample or per g of essential oil, while for hydrolates the results were expressed as GAE) per mL.

Discussion

-        The relevance and significance of the pharmacodynamic model applied in this research must be more described. It was previoully applied to evaluate the antimicrobial potential of the herbal preparations obtained from this species or other related species? Which are the strong points and the limitations of this model when it is used evaluate the antimicrobial potential of phytochemicals or phytopreparations?

ANSWER: Thank you for this observation. We added the explanation of using and impact of pharmacodinatic study for alternative antimicrobial substances. This frame was not conducted on S. montana samples earlier, so this is the first study which involved the comprehensive profilling of time-kill kinetic study with the obtained MIC values. We hope that this part of the study will be frame for further investigations of this plant conducted by other researchers.

Materials and methods:

-        Section 4.2.3: the extraction parameters need to be more described; can the authors mention the amplitude or/and the frequency used for develop UAE? Also, during UAE, the temperature was monitored, it was constant? It is well known that ultrasound treatment induces a thermal effect in the extraction mixture that influences the extraction yield and the final quality of the extract so, how it was monitored and quantified this effect?

ANSWER: Thank you for this suggestion, we added detail about the treatment in ultrasound bath. As we used low working temperature (which are favorable for the thermally unstable compounds), the thermal effect on the final quality of the extracts was not directly followed for this study.

-        Section 4.3. Could the authors bring a more appropriate chemical description for the SWE, UAE-MeOH, UAE-H2O, MAE-MeOH and MAE-H2O? As long as for the volatile compounds they provided an in-depth characterization of individual chemical components of volatile oils, for the uniformity of research design it could be added an more detailed description of phenolic profile for the afforementioned extracts.

ANSWER: The authors appreciate this suggestion. However, in this study we decided for characterization of volatile compounds only from essential oil and hydrolate by GC-MS. In future research we will include analysis of volatile compounds from extract, too. If you recommend, we will remove GC-MS profile from this manuscript because of uniformity of research.

-        Section 4.3: The analysis of TPC is not clearly described; the cited method is used for the assessment of aqueous solutions, hence, how the authors analyzed the volatile oils as long as this herbal preparations are not miscible with aqueous reagents used in this assay?  Also, it must be mentioned the standard compound used to express the results (I guess it is gallic acid based on the section ”Results”) This section must be improved.

ANSWER: Thank you! We noted that gallic acid is used as a standard compound to express the results.

-        Section 4.4: Same as previous, all these methods involve testing the antioxidant potential in aqueous solution. Please, provide a more detailed description of the methodology used in this section.

ANSWER: Thank you! We provided the explanation.

-        Section 4.5.4: Please, define the coeficients from the equation. Also, as I can see in the cited reference, the model is developed for describing the kinetics of an leavening process so, how this can be extrapolated to evaluate the pharmacodynamics of an antibacterial agent? Please, make an in-depth description of this method.

ANSWER: Thank you for this remark. The used reference (36) demostrated mathematical modelling approach, the reference (73) demostrated sustainable for biological systems, while the reference (72) represents direct implementation mathematical approach for pharmacodinamical potential of antimicrobials. In order to beter explain this approach, the following text was added to section 4.5.4:

...y(t) presents the bacterial and yeast growth in presence of MICs of antimicrobials (markers signify the experimental data; lines indicate predictive results), during the process.

The four parameter logistic regression function is suitable for biologic systems, which fits data to the S-shaped curves. The regression coefficients which participate in the function could be described as follows: a - the minimum value that was obtained (close to t=0), d - the maximum value that could be obtained (), c - the inflection point (the point on the S-shaped curve between a and d) and b = the Hill’s slope of the curve (the steepness of the curve at point c).

Reviewer 2 Report

The reviewed manuscript presents a comparative analysis of seven different fractions - herbal preparations, obtained from two taxa of the genus Satureja. The volatile compound profile of the essential oil and hydrolate fractions were characterized. Their biological activities were also demonstrated, as well as their total phenolic content. The phytochemical profile of a herbal preparation not only depends on the habitat from which the plant material was obtained, but also on the extraction method used. This is a very interesting aspect of the phytochemical analysis that the authors of the manuscript undertook, with regard to the possibility of using the obtained preparations for therapeutic purposes.

The research methodology and the presented results are correct. The authors should standardize the literature as required for the authors, e.g., item 1, 4, 14, 18, 21, 25, 26, 34, 37, 38, 39, 42, 45, 48, 56, 61, 62, 67. In Table 2, the names of the bacterial strains should be made clearer. In Table 3, EO - S.montana should be deleted from the row, as it is confusing, and aligned with the other extraction methods. Latin names in tables and figures should be italicized. Also, the name of the Salmonella strain should be corrected and standardized - whether Typhimurium or typhimurium.

Author Response

The reviewed manuscript presents a comparative analysis of seven different fractions - herbal preparations, obtained from two taxa of the genus Satureja. The volatile compound profile of the essential oil and hydrolate fractions were characterized. Their biological activities were also demonstrated, as well as their total phenolic content. The phytochemical profile of a herbal preparation not only depends on the habitat from which the plant material was obtained, but also on the extraction method used. This is a very interesting aspect of the phytochemical analysis that the authors of the manuscript undertook, with regard to the possibility of using the obtained preparations for therapeutic purposes.

The Authors would like to thank the Reviewer for a quick and professional review as well as the opportunity to make essential and crucial changes in our work. All the Reviewer' remarks are accepted and the paper is changed according to their comments. The Authors believe that the changed paper would satisfy the Reviewer' criteria and that it is going to be interesting enough for publishing in the Processes.

We decided to revise the manuscript according to the Reviewer' remarks, highlighting the changes directly in the revised manuscript.

The research methodology and the presented results are correct.

ANSWER: Thank you for this observation.

The authors should standardize the literature as required for the authors, e.g., item 1, 4, 14, 18, 21, 25, 26, 34, 37, 38, 39, 42, 45, 48, 56, 61, 62, 67.

ANSWER: Thank you for this observation, we corrected it.

In Table 2, the names of the bacterial strains should be made clearer.

ANSWER: Thank you for this observation, we expand the table.

In Table 3, EO - S.montana should be deleted from the row, as it is confusing, and aligned with the other extraction methods.

ANSWER: Thank you for this suggestion, we reorganized the table in order to simplify the results.

Latin names in tables and figures should be italicized. Also, the name of the Salmonella strain should be corrected and standardized - whether Typhimurium or typhimurium.

ANSWER: We checked all figures and tables, and corrected it.

Round 2

Reviewer 1 Report

In my opinion, the manuscript „Variability in biological activities of Satureja montana subsp.
montana and subsp. variegata based on different extraction methods” could be accepted in the actual format. Congratulations for your valuable work!